# Synthesis, Structure, and Magnetic and Biological Properties of Copper(II) Complexes with 1,3,4-Thiadiazole Derivatives

**DOI:** 10.3390/ijms241613024

**Published:** 2023-08-21

**Authors:** Lyudmila G. Lavrenova, Taisiya S. Sukhikh, Lyudmila A. Glinskaya, Svetlana V. Trubina, Valentina V. Zvereva, Alexander N. Lavrov, Lyubov S. Klyushova, Alexander V. Artem’ev

**Affiliations:** 1Nikolaev Institute of Inorganic Chemistry, Siberian Branch, Russian Academy of Sciences, Novosibirsk 630090, Russia; sukhikh@niic.nsc.ru (T.S.S.); glinsk@niic.nsc.ru (L.A.G.); svt@niic.nsc.ru (S.V.T.); zvereva@niic.nsc.ru (V.V.Z.); lavrov@niic.nsc.ru (A.N.L.); art@niic.nsc.ru (A.V.A.); 2Institute of Molecular Biology and Biophysics, Federal Research Center for Fundamental and Translational Medicine, Novosibirsk 630117, Russia; klyushovals@mail.ru

**Keywords:** coordination compounds, copper(II), 2,5-bis(ethylthio)-1,3,4-thiadiazole, 2,5-bis(pyridylmethylthio)-1,3,4-thiadiazole, synthesis, structure, biological and magnetic properties

## Abstract

New coordination compounds of copper(II) with 2,5-bis(ethylthio)-1,3,4-thiadiazole (L^1^) and 2,5-bis(pyridylmethylthio)-1,3,4-thiadiazole (L^2^) with compositions Cu(L^1^)_2_Br_2_, Cu(L^1^)(C_2_N_3_)_2_, Cu(L^2^)Cl_2_, and Cu(L^2^)Br_2_ were prepared. The complexes were identified and studied by CHN analysis, infrared (IR) spectroscopy, powder X-Ray diffraction (XRD), and static magnetic susceptibility. The crystal structures of Cu(II) complexes with L^1^ were determined. The structures of the coordination core of complexes Cu(L^2^)Cl_2_ and Cu(L^2^)Br_2_ were determined by Extended X-ray absorption fine structure (EXAFS) spectroscopy. Magnetization measurements have revealed various magnetic states in the studied complexes, ranging from an almost ideal paramagnet in Cu(L^1^)_2_Br_2_ to alternating-exchange antiferromagnetic chains in Cu(L^1^)(C_2_N_3_)_2_, where double dicyanamide bridges provide an unusually strong exchange interaction (J_1_/k_B_ ≈ −23.5 K; J_2_/k_B_ ≈ −20.2 K) between Cu(II) ions. The cytotoxic activity of copper(II) complexes with L^2^ was estimated on the human cell lines of breast adenocarcinoma (*MCF-7*) and hepatocellular carcinoma (*HepG2*).

## 1. Introduction

Polynitrogen-containing heterocycles are a promising class of ligands for the synthesis of 3d-block metal complexes that are strongly responsive to the changes in external conditions. Cu(II) complexes exhibit antiferro- or ferromagnetic exchange interactions between paramagnetic centres [1], while metal complexes with 3d*^n^* (*n* = 4–7) electron shells demonstrate spin crossover [2,3].

The synthesis and study of sulphur-containing azaheterocyclic compounds attract attention because of a wide range of biological activities demonstrated by these compounds, particularly 1,3,4-thiadiazole derivatives [4,5,6,7,8]. For example, they exhibit antibacterial, antifungal, anticancer, anticonvulsant, anti-inflammatory (COX-inhibitors), antitubercular, antiviral, and other types of the biological activity [8]. 1,3,4-Thiadiazole derivatives are also used as corrosion inhibitors in engineering [9] and in other fields. In this work, we synthesized and characterized copper(II) complexes with two 1,3,4-thiadiazole derivatives, 2,5-bis(ethylthio)-1,3,4-thiadiazole (L^1^) and 2,5-bis(pyridylmethylthio)-1,3,4-thiadiazole (L^2^) (Figure 1). The choice of these ligands was not arbitrary; their closest analogues attract interest as building blocks for a variety of molecular complexes and coordination polymers showing remarkable magnetic [10,11,12,13], bactericide [14], and optical [13,15] properties and interesting structural features [16,17,18,19]. Particularly, thiadiazoles can engage in specific intermolecular interactions, which is important for crystal engineering [20,21]. Therefore, the design of new metal derivatives based on 2,5-bis(organylthio)-1,3,4-thiadiazoles is a challenging task for modern coordination chemistry. No data on the complexes with such ligands have been yet reported.

## 2. Results

### 2.1. Synthesis and Characterization

Complexes Cu(L^1^)_2_Br_2_ (**1**), Cu(L^1^)(C_2_N_3_)_2_ (**2**), Cu(L^2^)Cl_2_ (**3**), and Cu(L^2^)Br_2_ (**4**) were isolated from ethanol solutions at various Cu(II) and L^1^, L^2^ ratios. The L:Cu(II) ratio was selected empirically to obtain a phase with a required composition (Figure 2). According to the powder XRD data, the resulting Cu(II) compounds with both L^1^ and L^2^ are crystalline, but we succeeded in growing single crystals only for copper(II) complexes with L^1^. The XRD-quality single crystals of complexes **1** and **2** with L^1^ precipitated from the mother liquor after the separation of the polycrystalline phase. The XRD patterns obtained for the polycrystalline phases of **1** and **2** agree well with the theoretical XRD patterns calculated for the structural models of **1** and **2** (Appendix A).

The IR spectra of L^1^, L^2^ and those of the complexes with these ligands contain ν(CH) vibrations at 3070–3000 cm^−1^ (Table 1) and bands of ν(CH_3_) and ν(CH_2_) vibrations of the L^1^, L^2^ ligands and the complexes at 3000–2850 cm^−1^. The region of 1700–1450 cm^−1^ contains coordination-sensitive bands of stretching and bending vibrations of 1,3,4-thiadiazole rings. The positions and number of these bands in the spectra of complexes with L^1^, L^2^ differ significantly from those of the ligands. This indicates that the nitrogen atoms of the heterocycle are coordinated to the copper(II) ion, which is also confirmed by the XRD data for complexes with L^1^. Both of the ligands and the complexes contain ν(S-C-S) bands in the region of 1090–1030 cm^−1^.

### 2.2. X-ray Structure Determination

The XRD patterns obtained for the polycrystalline phases of **1** and **2** agree well with the theoretical XRD patterns calculated for the structural models of **1** and **2** (Appendix A). According to the XRD data, the Cu(L^1^)_2_Br_2_ complex (**1**) is isostructural with its chloride analogue Cu(L^1^)_2_Cl_2_ that was prepared earlier [22]. It crystallizes in the triclinic crystal system, space group *P*–1, Z = 2.

Figure 1a shows the structure of complex **1**. The central Cu^2+^ ion coordinates nitrogen atoms of two monodentate L^1^ ligands and two bromide ions of the {CuN_2_Br_2_} coordination polyhedron. The bond lengths in CuN_2_Br_2_ agree well with the data from the CCDC structural database (version 5.43, November 2021) [23]. The bond angles at the Cu^2+^ ion vary from 91.98° to 158.30°, thereby indicating that the central atom occurs in a distorted square-planar environment with bromide ligands in trans positions (Figure 1b). The L^1^ ligands in **1** occur in trans positions and are rotated almost perpendicular to the plane of the coordination polyhedron. In the structure of the complex, the atoms of one of the two thiolate groups are disordered over two positions S3A and S3A′.

Single crystals of Cu(L^2^)(C_2_N_3_)_2_ (**2**) also belong to the triclinic crystal system, space group *P*–1, Z = 2, (Appendix A). The structure is composed of polymeric 1D chains with repeating {Cu(L^2^)(μ-(C_2_N_3_)_2_} fragments (Figure 2). The Cu^2+^ coordination sphere contains four N atoms of two equatorial dicyanamide ions with the Cu-N distances varying from 1.974(2) to 1.981(1) Å, and an apical N atom of the thiadiazole ring (the Cu-N distance is 2.243(2) Å). The average deviation of nitrogen atoms from the basal plane is 0.02 Å. The N atoms form a tetragonal pyramid, with the angle between the Cu1-N2 vector and the basal plane equal to 87.8°. Two neighbouring copper ions are connected by two bridging C_2_N_3_^–^ ions forming zigzag chains (Figure 3a,b). Each {Cu_2_(μ-C_2_N_3_)_2_} metallocycle in these polymer chains adopts a chair conformation with the Cu1 and Cu1′ atoms deviating from the plane by −0.567 Å and 0.567 Å, respectively.

It is worth emphasizing that the Cu…Cu distances inside each chain alternate and are equal to 6.950 Å and 7.050 Å (Figure 2). In turn, the distances between the chains are significantly larger, the shortest of them being equal to 7.840 Å (Figure 3b). There are also S…N contacts between the chains: the shortest of them is the contact between the sulphur atom of the thiadiazole ring and the dicyanamide nitrogen atom (S2…N4 = 3.178(2) Å) (shown by dashed lines in Figure 3a), which is shorter than the sum of their van der Waals radii (3.45 Å).

### 2.3. EXAFS Spectroscopy Determination

A structure of the coordination cores in complexes **3** and **4** was determined by EXAFS spectroscopy. The XANES and EXAFS spectra obtained for these compounds are shown in Figure 4a,b.

Figure 5 and Figure 6 show the k^2^-weight EXAFS spectra fitted with the EXCURVE program [24], with the Δk ranging from 3 to 11 Å^−1^. The initial models of the EXAFS spectra were based on the XRD data for the mononuclear Cu(L^1^)_2_Br_2_ complex and cupric chloride complex representing a 1D coordination polymer, which was formed at the expense of the triple bridge between two neighbouring copper(II) cations (double chlorido bridge and the tetrazole ring N3,N4-bridge) [25]. EXAFS fitting results are presented in Table 2 for complex **3** and in Table 3 for **4**. Note that the EXAFS allow us to determine average coordination numbers, interatomic distances, and Debye–Waller factors for the scattering atom (Cu). The parameters were determined from the EXAFS data with an accuracy of ±1% for interatomic distances, ±10% for coordination numbers, and ±20% for Debye–Waller factors.

### 2.4. Magnetic Properties

For all the complexes under investigation, the dependences of the molar magnetic susceptibility χ on temperature and magnetic field were measured in the ranges T = 1.77–330 K and H = 0–10 kOe. The data obtained under zero-field cooled and field-cooled conditions revealed perfect magneto–thermal reversibility in all samples, which pointed to the absence of any ferromagnetic or spin-freezing phenomena and allowed a straightforward analysis of the χ(T,H) dependences. After subtracting the diamagnetic core contribution, the remaining paramagnetic component of the magnetic susceptibility, χ_p_(T), was analysed using the Curie–Weiss dependence $\chi_{p}\left(T\right)=N_{A}\mu_{\mathrm{eff}}^{2}/3k_{B}\left(T-\theta \right)$ to obtain insights into the spin states of Cu ions and the exchange interactions between them. 

The simplest magnetic behaviour, barely distinguishable from that of an ideal paramagnet, has been found for complex **1** (Figure 7). Indeed, the measured (1/χ_p_)(T) curve of **1** turns out to be almost perfectly linear and goes straight to the origin, tending to intersect the temperature axis at |θ| less than 0.05 K. Given the small value of θ, we can calculate the effective magnetic moment μ_eff_ neglecting the exchange interactions between copper ions. The resulting μ_eff_ amounts to ≈ 1.83 μ_B_ at T = 300 K, which is close to the spin-only value of 1.73 μ_B_ expected for Cu^2+^ (S = 1/2) ions, with a slight excess being caused by the contribution from orbital moments. As can be seen in Figure 7, μ_eff_ measured at H = 1 kOe decreases smoothly to μ_eff_ ≈ 1.80 μ_B_ at T = 20 K and somewhat faster at lower temperatures down to μ_eff_ ≈ 1.78 μ_B_ at T = 1.77 K; the latter implies the presence of a tiny, barely detectable antiferromagnetic (AF) interaction between Cu^2+^ ions. 

A noticeably stronger AF exchange interaction is observed in complexes **3** and **4** (Figure 8a,b). Curie–Weiss fitting of the high-temperature χ_p_(T) data results in the θ values of ≈−6.4 K and ≈−5.9 K for **3** and **4**, respectively. When we select for fitting the low-temperature region, where the χ_p_ data are much less sensitive to such uncertain parameters as the Van Vleck paramagnetism of Cu^2+^ ions and the temperature-dependent orbital contribution, the Weiss constant θ diminishes to −0.70 K and −0.64 K, yet still remains sizable. The evaluated effective magnetic moment of **3** reaches μ_eff_ ≈ 1.87 μ_B_ at T = 300 K (μ_eff_ ≈ 1.86 μ_B_ for θ ≈ −0.70 K), which is typical for Cu^2+^ (*S* = 1/2) ions, and slightly exceeds the spin-only value of ≈1.73 μ_B_ (Figure 8a). Although the shape of the μ_eff_(T) curve in **4** is remarkably similar to that of **3**, the absolute μ_eff_ values turn out to be noticeably lower (Figure 8b). The latter implies an admixture of a non-magnetic impurity phase in the studied sample of **4**.

The most interesting magnetic behaviour was revealed for complex **2** (Figure 9a,b). In contrast to the aforementioned samples, **2** demonstrates a non-monotonic χ_p_(T) dependence that passes through a broad peak at T_max_ ≈ 10.45 K and then recovers a steep growth upon cooling below T ≈ 4.5 K (Figure 9a). Such a broad magnetic susceptibility peak is a fingerprint of low-dimensional antiferromagnetic systems, such as dimers or 1D chains [26,27,28,29,30], where the AF exchange interactions are capable of providing local spin correlations, but the absence of 3D links prevents a phase transition to the long-range ordered AF state. In its turn, the low-temperature χ_p_ growth is often observed in such systems owing to the presence of monomers or chain fragments with an odd number of ions, which behave as isolated paramagnetic centres. The Curie–Weiss fitting of χ_p_(T) data in the high-temperature region of 40–300 K, i.e., far above T_max_, results in the θ value of ≈−16 K and μ_eff_ ≈ 1.91–1.92 μ_B_ (Figure 9b). A detailed analysis of the observed magnetic behaviour side-by-side with structural data will be given in Section 3.

### 2.5. In Vitro Cytotoxic Activity

The effect of the novel compounds on the viability of *MCF-7* and *HepG2* cells was evaluated using the Hoechst 33342/PI dual staining method and subsequent differentiation of the cells into live, dead, and apoptotic ones. The half maximal inhibitory concentration IC_50_ was defined as the concentration of the substance at which the percentage of live cells was equal to 50%. To determine the IC_50_ value, experimental dependences of cell survival (%) on the substance concentration (μM) were fitted by nonlinear functions.

The copper(II) halide complexes Cu(L^2^)Cl_2_ and Cu(L^2^)Br_2_ did not affect the viability of *HepG2* cells (Figure 10a,b, respectively), but had a dose-dependent cytotoxic effect on *MCF-7* cells after a 48 h period of incubation (Figure 10c,d, respectively). The IC_50_ values characterizing the cytotoxic activity of the complexes on the *MCF-7* line are 41.7 ± 0.4 and 39.0 ± 0.4 μM, respectively. Both compounds exhibit similar cytotoxic effects. Thus, the cytotoxic activity of the copper(II) complex with L^2^ does not depend on the halide ion in its composition. At the same time, it is worth emphasizing that the studied complexes selectively affect the cell lines of different origins.

## 3. Discussion

### 3.1. Structure of Complexes ***1*** and ***2***

Literature review reveals that, similar to **1** complexes, Cu(tda)_2_Hal_2_ (Hal = Cl, Br) (tda = 2,5-bis(methylthio)-1,3,4-thiadiazole), whose ligand contains a thiomethylate group instead of a thioethylate group, are centrosymmetric, i.e., their Hal–Cu–Hal and N–Cu–N angles are strictly equal to 180° [30]. Moreover, the thiadiazole ligands in the Cu(tda)_2_Hal_2_ complexes are rotated by 180° relative to the N–Cu–N line (Figure 11). In the present case of complex **1**, the ligands are located opposite to each other (the corresponding angle is ~4°). Different geometries of molecules cause the crystal packing in the complexes to be also different; namely, Cu(tda)_2_Hal_2_ contains intermolecular Hal⋯S contacts (Hal = Cl, Br), while the distances in **1** are significantly larger (the minimal distance in **1** is 3.74 Å, Figure 1b).

The CCDC database contains no structures of 1D polymer chains with repeating {Cu_2_(μ-C_2_N_3_)_2_} fragments where the copper(II) coordination number would be equal to 5. At the same time, 12 discrete binuclear copper complexes with such fragments were found. For instance, in the structure of one of these complexes, [Cu_2_(dmphen)_2_(C_2_N_3_)_4_] [31], the Cu-Cu distance within dimers is 7.134(1) Å, while the shortest distance between dimers is 6.752(1) Å. In the [Cu_2_(medpt)_2_(C_2_N_3_)_2_](ClO_4_)_2_ binuclear complex [32], the Cu-Cu distance is 6.975 Å within dimers and as large as 8.562 Å between them. In the layered polymer Cu(L_2_)_2_(C_2_N_3_)_2_ studied in our earlier work [33], the distances between copper ions connected by cyanamide bridges are 7.30 Å and 7.707 Å.

The coordination environment of Cu(II) depends on the nature of ligands (the number of donor atoms, charge, skeletal rigidity, etc.), reaction conditions, and the presence of secondary bonding interactions forming a specific crystal packing. These factors cumulatively affect the organization of a coordination polyhedron, sometimes in unpredictable ways. In a recent paper [22], we have shown that CuCl_2_ and L^1^ can form either tetra- or penta-coordinated complexes upon slight variation of the reaction conditions. Penta- and hexa-coordinated complexes [Cu(tda)(C_2_N_3_)_2_] and [Cu(tda)_2_(C_2_N_3_)_2_] are also formed in similar conditions [30]. To obtain an idea for which coordination number is preferred, we performed a CSD survey of crystal structures comprising a fragment {CuL_x_Br_y_}, where L = κ-N heterocyclic ligand with the N having three neighbours (i.e., the N is sp^2^-hybridized), x, y = 2–4 (Appendix A). Tetra-coordinated fragment {CuL_2_Br_2_} has 268 hits, while penta-coordinated ones have a total of 124 hits, suggesting the former fragment is noticeably more common. Note that hexa-coordinated fragments have only a total of 39 hits; this is likely a consequence of high steric hindrance of heavy Br atoms. Complexes with dicyanamide are less common than bromide ones. The CSD survey of structures comprising at least one dicyanamide coordinated to Cu revealed that penta-coordinated complexes (61 hit) dominate over tetra- (10 hits) and hexa-coordinated ones (24 hits; Appendix A). Thus, the Cu polyhedra in structures **1** and **2** belong to the most common coordination environment: tetracoordinated for the bromide, and pentacoordinated for the dicyanamide.

### 3.2. Structure of Complexes ***3*** and ***4*** According to the EXAFS Data

The experimental XANES and EXAFS spectra (Figure 4a,b) of Cu(L^2^)Cl_2_ (**3**) and Cu(L^2^)Br_2_ (**4**) differ significantly, thus indicating that Cu(II) ions occur in different coordination in these compounds. The simulations of experimental EXAFS spectra (Figure 5 and Figure 6) have shown that complex **3** is well described by the trimer structure (Figure 12a), while **4** conforms to the dimer one (Figure 12b).

The fitting results of EXAFS spectra for **3** signify the formation of a trimer, as follows from the obtained coordination numbers (C.N.) and the interatomic distances (Table 2). Copper(II) ions form three different coordination sites: CuNCl_3_, CuN_2_Cl_3_, and CuN_2_Cl_4_ (Figure 12a). The EXAFS spectrum of the complex reveals two different average values for Cu-Cl′ (2.26 Å) and Cu-Cl″ (2.67 Å) distances with the corresponding average C.N. values equal to 2.6 and 0.8, respectively. The total coordination number of Cu(II) is 3.4, which is very close to the average C.N. value of Cu(II) in the case of trimers (3.3(3)). Each Cu(II) ion coordinates nearest nitrogen atoms (N′) at an average distance of 1.97 Å (Cu-N′) and remote nitrogen atoms (N″) at an average distance of 3.00 Å (Cu-N″) (Table 2). In the trimer model, the average C.N. value of Cu(II) should be ~1.67, both for the nearest and the remote nitrogen atoms. Thus, the C.N. values of 1.6 obtained from the fitting of EXAFS spectra both for Cu-N′ and Cu-N″ (Table 2) agree very well with the trinuclear structure of the Cu(L^2^)Cl_2_ complex.

According to EXAFS spectra fitting for **4**, the dimer is the most likely model (Figure 12b). Each Cu(II) ion in the dimer coordinates two nitrogen atoms N′ + N″ with distances 2.00 Å and 2.94 Å, respectively (Table 3). The coordination is supplemented by two bromide ions (Figure 12b). The average Cu-Br distance is 2.43 Å, which is in good agreement with the XRD data for the Cu(L^1^)_2_Br_2_ (**1**) mononuclear complex. 

### 3.3. Magnetic Properties of the Complexes

With the structural data in hand, we can discuss the mechanisms of interactions between Cu^2+^ ions in the studied complexes. In the case of Cu(L^1^)_2_Br_2_ (**1**), the situation is quite simple: the large distance between Cu^2+^ ions in its crystal structure and the absence of pathways capable of providing exchange interaction leave the metal ions almost perfectly isolated from each other. The observed AF interaction is so weak (|θ| < 0.05 K) that one can hardly conclude with certainty whether it is of exchange or dipole–dipole origin. A bit stronger yet still very weak (θ ≈ −0.2 K) AF interaction was recently reported for the isostructural Cu(L^1^)_2_Cl_2_ complex [22], confirming the magnetically dilute state of Cu^2+^ ions in this family of compounds.

In contrast to **1**, the explanation of magnetic properties of **3** and **4** (Figure 8) is challenging. According to the EXAFS data, these complexes possess remarkably different structures, with Cu^2+^ ions in **3** and **4** being arranged in trimer and dimer blocks, respectively. Given the short distance between Cu^2+^ ions in the crystal structure (≈3.3 Å in **3**, and ≈3.89 Å in **4**), one would naturally expect a pronounced behaviour of magnetic trimers in **3** and dimers in **4**. In the former case, the effective magnetic moment should decrease by √3 times with cooling, since AF trimers behave at low temperatures (k_B_T << J) essentially as paramagnetic S = 1/2 centres. In the latter case, the magnetic susceptibility should pass through a maximum at a certain temperature and then drop to zero in the zero-*T* limit, where AF dimers turn into the ground-state singlet state [26,34]. Apparently, these expectations fail; the χ_p_(T) data (Figure 8) for neither **3** nor for **4** follow the expressions developed for AF dimers and trimers [26,34]. To make a side-by-side comparison of complexes **3** and **4** whose magnetic behaviours should allegedly be very different, we plotted their μ_eff_(T) dependences in Figure 13. As can be seen, the μ_eff_(T) curves of **3** and **4** turn out to be surprisingly similar and resemble the behaviour of a uniform isotropic antiferromagnet, with just the exchange interaction in **4** being slightly (by ~10%) weaker than in **3**. Additional information can be obtained from the low-T M(H) data that should follow the theoretical dependence $M\left(H\right)=N_{A}{g\mu }_{B}{\mathrm{SB}}_{S}\left(\frac{{g\mu }_{B}}{k_{B}T}\mathrm{SH}\right)$ based on the Brillouin function *B_S_*(*x*). A fit taking into account the isotropic AF exchange interaction (θ ≈ −0.70 K for **3**) and the g-factor of Cu^2+^ ions g ≈ 2.20 gives a pretty good description of the M(H) data for **3** (inset in Figure 13) without any additional adjustable parameters. This gives a solid evidence that the magnetic system of **3** is composed of paramagnetic S = 1/2 centres whose number is equal to the number of Cu^2+^ ions, but not of trimers. The same is apparently true for the complex **4**. The only apparent way to reconcile the observations that the Cu^2+^ ions in **3** and **4** act magnetically as individual centres, while structurally they are arranged in trimers and dimers, is to assume that the bonds inside trimers and dimers are ineffective in delivering the exchange interaction. The latter turns the dinuclear and trinuclear structural blocks into sets of individual Cu^2+^ ions that interact with ions from neighbouring molecules. 

At a first glance, the complex Cu(ettda)(C_2_N_3_)_2_ (**2**) exhibits a rather ordinary magnetic behaviour of AF chains of Cu^2+^ ions supplemented by a Curie–Wiess contribution χ_CW_(T) from monomers and chain fragments consisting of an odd number of Cu^2+^ ions [27,28,29,30,35]. A common approach to fit the magnetic susceptibility χ_ch_(T) from uniform AF Heisenberg chains described by the Heisenberg Hamiltonian $H=-J\sum_{i}\;{\vec{S}}_{i}\;\cdot \;{\vec{S}}_{i+1}$ (J < 0 is for the AF exchange) is to use numerical calculations [27,29,35] or a well-known polynomial approximation [28]. A fine tuning of the model fit is usually achieved by introducing an additional inter-chain interaction [28,29,30] or by modifying J or the g-factor [29,30]. However, these approaches failed to produce a fair-quality fit to the χ_p_(T) data of **2**, and so did the dimer model [26]. 

To make a more detailed analysis of the χ_p_(T) data of **2**, one needs, firstly, to separate the contributions from Cu^2+^ chains, χ_ch_(T), and from individual paramagnetic (PM) centres, χ_CW_(T). This can be performed by examining the low-T magnetization curve M(H) (Figure 14a), since the chain susceptibility at available fields should be virtually independent of H, while the magnetization of PM centres follows the dependence $M\left(H\right)=N_{A}{g\mu }_{B}{\mathrm{SB}}_{S}\left(\frac{{g\mu }_{B}}{k_{B}T}\mathrm{SH}\right)$. As can be seen in Figure 14a, the measured χ(H)/χ(0) is very close to the behaviour of isolated S = 1/2 centres, implying that it is the PM centres that dominate the low-T magnetization. Calculations show that 5.9% of Cu^2+^ ions in the sample **2** behave as isolated S = 1/2 centres; the evaluated χ_CW_(T) curve is shown by the solid line in Figure 14b.

The χ_ch_(T) contribution (red symbols in Figure 14b) obtained by subtracting χ_CW_(T) from the raw χ_p_(T) data passes through a maximum at T′_max_ ≈ 14.1 K with a subsequent decrease to zero with further cooling. This behaviour is much different from that of uniform AF S = 1/2 chains, where the magnetic susceptibility drops by only one quarter in the zero-T limit, and points to the formation of a spin gap in the Cu^2+^ chains in the complex **2**. Indeed, the lowest temperature part of the χ_ch_(T) curve can be well described by a simple expression for chains with a spin gap: χΔ(T) = aT^−0.5^exp(−Δ/k_B_T) [29,36], with a spin gap Δ/k_B_ ≈ 7 K (inset in Figure 14b). Furthermore, the χ_ch_(T) data in a wide temperature range can be well fitted by a polynomial expression [37], χ_alt_(T), suggested for alternating-exchange AF chains described by the Hamiltonian $H=-J\sum_{i}^{n/2}\left(S_{2i-1}\;\cdot \;{\vec{S}}_{2i}+\delta {\vec{S}}_{2i}\;\cdot \;{\vec{S}}_{2i+1}\right)$ [28,29,36,37] (dashed blue lines in Figure 14b). The best fit was obtained for the alternation parameter δ = 0.86 and the largest exchange integral J/k_B_ = −23.5 K, which means that the exchange interaction J/k_B_ along the Cu^2+^ chains in **2** alternates from −23.5 K to −20.2 K. It is worth noting that, according to numerical calculations of Ref. [29], a spin gap emerging in a partially dimerized S = 1/2 AF chain with J/k_B_ = −23.5 K and δ = 0.86 should amount to Δ/k_B_ ≈ 6.3 K, which agrees well with our fitting result of Δ/k_B_ ≈ 7 K. 

As a matter of fact, the formation of alternating AF chains in the complex **2** could be expected straight from its structural data, which show that the distance between Cu^2+^ ions along the chain alternates from 6.95 Å to 7.05 Å (Figure 2). A surprising feature that could hardly be expected is the remarkably strong exchange interaction found for dicyanamide-bridged Cu^2+^ ions in **2**. Usually, the ability of dicyanamide bridges to transfer exchange interactions between copper ions is considered to be very poor, resulting in the J/k_B_ values of about 1 K or below [31,32]. To the best of our knowledge, the strongest coupling of Cu^2+^ ions double-bridged by dicyanamide was found for [Cu_2_(dmphen)_2_(C_2_N_3_)_4_] (dmphen = 2,9-dimethyl-1,10-phenanthroline) with J/k_B_ ≈ 4.75 K, where the susceptibility exhibited maximum at T_max_ ≈ 3 K [31]; this was suggested to represent the upper limit of the AF interaction through dicyanamide bridges. The present work demonstrates the ability of dicyanamide to deliver an almost five times stronger exchange interaction (J/k_B_ ≈ −23.5 K; T′_max_ ≈ 14.1 K).

### 3.4. Cytotoxic Properties of the Complexes

The study of cytotoxic effects of new copper(II) halide complexes with 2,5-bis(pyridylmethylthio)-1,3,4-thiadiazole showed that complexes Cu(L^2^)Cl_2_ (**3**) and Cu(L^2^)Br_2_ (**4**) have no effect on the viability of *HepG2* hepatocellular carcinoma cells in the concentration range from 1 to 50 μM after a 48 h exposure. The *HepG2* line is quite commonly used for the early in vitro assessment of potential hepatotoxicity of novel molecules [38]. Note, however, that the activity and expression of some xenobiotic-metabolizing enzymes in these cells are significantly lower than those of non-tumour human liver samples [39,40,41]. For example, *CYP2C9*, *CYP2C19*, and *CYP3A4*, which are mainly located in the liver, are diagnostic markers of hepatocellular carcinoma [42,43,44]. Therefore, not all toxic effects of reactive metabolites can be correctly estimated on *HepG2*-based models. Under experimental conditions similar to those of the present work, classical medications carboplatin and cisplatin had a more significant effect on *HepG2* cells than the new complexes (Table 4).

The present study also showed that new copper(II) halide complexes have different effects on the cell lines of different origins: in the studied concentration range, the compounds have a cytotoxic effect on the *MCF-7* breast adenocarcinoma cells but did not cause the death of the *HepG2* cells. Since no difference between the effects of **3** and **4** on the cells was revealed, we conclude that their cytotoxic activity does not depend on the halide ion. Under similar experimental conditions, the IC_50_ values (characterizing the cytotoxic activity) of carboplatin and cisplatin with respect to *MCF-7* cells are similar to those of **3** and **4** (Table 4).

The in vitro cytotoxicity study demonstrated that the further search for complexes exhibiting cytotoxic and potential antitumor activity in this series of compounds is a promising direction of research.

## 4. Materials and Methods

All of the reagents and solvents were obtained from commercial sources and used without purification. The CHN analysis was performed at the analytical laboratory of the Nikolaev Institute of Inorganic Chemistry SB RAS using a EuroVector EURO EA 3000 analyser (Pavia, Italy). The IR absorption spectra were registered using a Scimitar FTS 2000 spectrometer in the range from 4000 to 400 cm^−1^. The studied samples were prepared in the form of suspensions in nujol and in fluorinated oils.

### 4.1. Synthesis

#### 4.1.1. Synthesis of the Ligands

2,5-Bis(ethylthio)-1,3,4-thiadiazole (L^1^) and 2,5-bis(pyridylmethylthio)-1,3,4-thiadiazole (L^2^) were synthesized according to the procedures reported in [18,47]. 

#### 4.1.2. Synthesis of Cu(L^1^)_2_Br_2_ (**1**)

Weighted samples of CuBr_2_ (1 mmol, 0.22 g) and L^1^ (1 mmol, 0.21 g) were dissolved by heating in ethanol (10 mL). The obtained brown solution was evaporated and cooled in a crystallizer with ice. After the settling, XRD-quality brown crystals precipitated. The crystals were filtered off, washed doubly with small portions of ethanol, and dried in air. The yield of XRD-quality crystals of **1** was 70%. All of the subsequent studies were carried out for the single-crystal phase. 

Found (%): C, 22.6; H, 3.0; N, 8.7.

Calculated for C_12_H_20_Br_2_CuN_4_S_6_ (%): C, 22.7; H, 3.2; N, 8.8.

#### 4.1.3. Synthesis of Cu(L^1^)(C_2_N_3_)_2_ (**2**)

Weighted samples of Cu(NO_3_)_2_·3H_2_O (2 mmol, 0.48 g) and L^1^ (4 mmol, 0.84 g) were dissolved by heating in ethanol (10 mL). The solutions were mixed, and a solution of NaC_2_N_3_ (0.28 g) in water (5 mL) was added to the mixture. The resulting cyan solution was evaporated and cooled in a crystallizer with ice. After the settling, XRD-quality blue-grey crystals precipitated. The crystals were filtered off, washed doubly with small portions of ethanol, and dried in air. The yield of complex **2** was 11%. 

Found (%): C, 29.7; H, 2.4; N, 27.8.

For C_10_H_10_CuN_8_S_3_ calculated (%): C, 29.9; H, 2.5; N, 27.9.

#### 4.1.4. Synthesis of Cu(L^2^)Cl_2_ (**3**), Cu(L^2^)Br_2_ (**4**)

Weighted samples of CuCl_2_·2H_2_O (0.5 mmol, 0.09 g) or CuBr_2_ (0.5 mmol, 0.11 g) salts and L^2^ (1.25 mmol, 0.28 g) were dissolved separately by heating in ethanol (10 mL), and the solutions were mixed. After removing the excess solvent (~1/3 of the volume) and cooling the solution, a precipitate was formed. The precipitate was filtered off, washed several times with ethanol, and dried in air. The yield of complexes **3** and **4** was 57% and 89%, respectively.

Found (%): C, 35.4; H, 2.7; N, 11.5.

For C_14_H_12_Cl_2_CuN_4_S_3_ calculated (%): C, 36.0; H, 2.6; N, 12.0.

Found (%): C, 30.5; H, 2.2; N, 10.1.

For C_14_H_12_Br_2_CuN_4_S_3_ calculated (%): C, 30.3; H, 2.8; N, 10.1.

### 4.2. XRD

The powder XRD patterns of polycrystalline samples 1 and 2 were collected with a BrukerD8 Advance diffractometer (sealed 40 kV/40 mA ceramic tube, LYNXEYE XE T detector, energy-discriminated CuK_α_) in the Bragg—Brentano geometry at room temperature. The samples were ground in heptane and deposited in the form of a ~0.1 mm thick layer on a plastic substrate. The experiment was conducted in the range of 5–70° with a step of 0.03° and the total counting time of 96 s/point. 

The single-crystal XRD data for the crystals of **1** and **2** were collected at 150 K with a Bruker D8 Venture diffractometer (ω- and φ-scans with a step of 0.5°, three-circle fixed-χ goniometer, CMOS PHOTON III detector, Mo-I µS 3.0 microfocus source, focusing Montel mirrors, λ = 0.71073 Å, MoKα radiation, N_2_-flow thermostat). The crystal structures were solved using the ShelXT [48] and refined using the ShelXL [49] programs assisted by Olex2 GUI [50]. Hydrogen atoms were located geometrically and refined in the riding model. Atomic displacements for non-hydrogen atoms were refined in harmonic anisotropic approximation with the exception of atoms of disordered thiolate groups in **1** belonging to a minor position. The corresponding S–C and C–C bond distances were restrained to be similar for the major and minor parts. The structures of **1** and **2** were deposited to the Cambridge Crystallographic Data Centre (CCDC) as a supplementary publication (No. 2259018 and 2259019). More detailed data were deposited to the CCDC and can be obtained at https://www.ccdc.cam.ac.uk/structures/ (accessed on 16 August 2023). The main crystal data and refinement details for **1** and **2** are summarized in Appendix A. The main interatomic distances and bond angles are listed in Appendix A.

### 4.3. X-ray Absorption Spectroscopy

The X-ray absorption spectra (XAS) were recorded in the 8 beamline channel at the Siberian Synchrotron and Terahertz Radiation Center located in the Novosibirsk VEPP-3 storage ring at Budker Institute of Nuclear Physics SB RAS [51]. The XAS spectra of the complexes were registered in the range from 150 eV, before the Cu*K* edge, to 800 eV, after the Cu*K* edge, in a standard transmission mode using ionization chambers filled with Ar/He and Xe serving as the monitoring and final detectors, respectively. A Si(111) slotted single crystal was used as a two-crystal monochromator. The storage ring operated with an energy of 2 GeV and a current of 70–140 mA. To perform the measurements, the complexes were mixed with a cellulose powder as a filler and pressed into pellets. The EXAFS data extraction (pre-edge subtraction, spline background removal) was performed using a VIPER 10.17 software package [52].

### 4.4. Magnetic Susceptibility

Magnetization measurements were carried out using a Quantum Design MPMS-XL SQUID magnetometer in the temperature range 1.77–300 K at magnetic fields H up to 10 kOe. To test the thermomagnetic reversibility, temperature dependences of the magnetization, M(T), were measured on heating the sample after it had been cooled either in zero magnetic field or in a given magnetic field, as well as upon cooling the sample. In order to determine the paramagnetic component of the molar magnetic susceptibility, χ_p_(T), the temperature-independent diamagnetic contribution, χ_d_, and a possible magnetization of ferromagnetic micro-impurities, χ_FM_(T) were evaluated and subtracted from the measured values of the total molar susceptibility χ = M/H. While χ_d_ was calculated using the Pascal’s additive scheme, χ_FM_(T), if any, was determined from the measured isothermal M(H) dependencies and the M(T) data taken at different magnetic fields. To determine the effective magnetic moment, µ_eff_, and the Weiss constant, θ, the paramagnetic susceptibility χ_p_(T) was analysed using the Curie–Weiss dependence $\chi_{p}\left(T\right)=N_{A}\mu_{\mathrm{eff}}^{2}/3k_{B}\left(T-\theta \right)$, where N_A_ and k_B_ are the Avogadro and Boltzmann numbers, respectively.

### 4.5. In Vitro Cytotoxicity Assay

The cytotoxic and cytostatic effects of the compounds were tested by the Hoechst/propidium iodide (PI) double staining protocol [45] on the MCF-7 (breast adenocarcinoma) and HepG2 (hepatocellular carcinoma) human cell lines. The lines were provided by our colleagues from the State Research Center of Virology and Biotechnology VECTOR. The cells were cultured in Iscove’s Modified Dulbecco’s Medium (IMDM, pH 7.4) with a 10% content of foetal bovine serum (FBS, HyClone, Sigma-Aldrich, Buchs, Switzerland) in a CO_2_ incubator (humidified atmosphere, 5% CO_2_ content, 37 °C) for 24 h after they were seeded in 96-well plates (5 × 10^3^ cells per well). After a 48 h period of incubation with the tested compounds (1–50 µM concentration; solvent concentration in the culture medium <1%), the cells were stained with a mixture of Hoechst 33342 (Sigma-Aldrich, Buchs, Switzerland) and PI (Invitrogen, Inchinnan, UK) fluorescent dyes for 30 min at 37 °C. The images (4 fields per well, 200× magnification) acquired on an IN CellAnalyzer 2200 system (GE Healthcare, Chalfont Saint Giles, UK) were analysed using the In Cell Investigator software (version 1.5, GE Healthcare, Chalfont Saint Giles, UK). The cytotoxic activity was estimated from the half-maximal inhibitory concentration (IC_50_) that was calculated from the dependence of the number of live cells (%) on the concentration of the test compound (μM).

## 5. Conclusions

In the present work, a number of new copper(II) complexes with 2,5-bis(ethylthio)-1,3,4-thiadiazole (L^1^) and 2,5-bis(pyridylmethylthio)-1,3,4-thiadiazole (L^2^) with compositions Cu(L^1^)_2_Br_2_, Cu(L^1^)(C_2_N_3_)_2_, Cu(L^2^)Cl_2_, and Cu(L^2^)Br_2_ were prepared and comprehensively studied. According to magnetic measurements, dicyanamide bridges in Cu(L^1^)(C_2_N_3_)_2_ provide an unexpectedly strong exchange interaction between Cu^2+^ ions, linking them into alternating-exchange antiferromagnetic (AF) chains, while, in the other complexes, only weak isotropic AF interaction is observed, at most. The cytotoxic activity of copper(II) complexes with L^2^ was tested on breast adenocarcinoma (MCF-7) and hepatocellular carcinoma (HepG2) human cell lines. The results indicate that this class of compounds holds promise as potential antitumor drugs.

## Data Availability

Not applicable.

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
