# Peer review of "Synthesis, Structure, and Magnetic and Biological Properties of Copper(II) Complexes with 1,3,4-Thiadiazole Derivatives"

_ijms, 2023, doi:10.3390/ijms241613024_

Round 1
Reviewer 1 Report
This is an important work from Lavrenova's school which is well known in coordination chemistry, and which provides a master piece with this work containing synthesis, crystal structures, magnetic and biological properties. Worth to say that the magnetic data are precisely discussed and investigated in depth. The readers of IJMS will thus enjoy this contribution which is expected to attract a high number of citations in the field of thiadazole complex derivatives.
Author Response
We thank the referee for a careful reading of the article and a high appreciation of our work.
Reviewer 2 Report
Lavrenova et al. investigated copper complexes of two 1,3,4-thiadiazole based motifs. Crystal structures with two different counter ions are studied. The authors have effective calculated magnetic moment and magnetic susceptibility. Metal complexes also exhibited cytotoxicity in MCF-7 cell line. The manuscript is very much suitable to be published in the journal ‘’IJMS’’.
1. One copper complex displayed tetra-coordinated environment whereas other one exhibited penta-coordinated arrangement surrounding copper. It will be more engaging if the authors would provide the rationale behind such type of structural variation. Recent literature such as Pesticide biochemistry and physiology 2017, 143, 26-32; Polyhedron 171 (2019) 559–570 should be provided. Importance of intermolecular interactions in structural stability should be addressed in the introduction.
2. Similar kinds of interesting complexes have already been reported by the same group. The authors need to provide synthetic scheme in the manuscript.
3. Are these complexes exhibit fluorescence? It would better if UV –visible spectra of these complexes are highlighted with different transitions.
Lavrenova et al. investigated copper complexes of two 1,3,4-thiadiazole based motifs. Crystal structures with two different counter ions are studied. The authors have effective calculated magnetic moment and magnetic susceptibility. Metal complexes also exhibited cytotoxicity in MCF-7 cell line. The manuscript is very much suitable to be published in the journal ‘’IJMS’’.
1. One copper complex displayed tetra-coordinated environment whereas other one exhibited penta-coordinated arrangement surrounding copper. It will be more engaging if the authors would provide the rationale behind such type of structural variation. Recent literature such as Pesticide biochemistry and physiology 2017, 143, 26-32; Polyhedron 171 (2019) 559–570 should be provided. Importance of intermolecular interactions in structural stability should be addressed in the introduction.
2. Similar kinds of interesting complexes have already been reported by the same group. The authors need to provide synthetic scheme in the manuscript.
3. Are these complexes exhibit fluorescence? It would better if UV –visible spectra of these complexes are highlighted with different transitions.
Author Response
We thank the referee for careful reading of the article and comments. Please find attached the responses.

Reviewer 3 Report
The authors have reported “Synthesis, structure, magnetic and biological properties of copper(II) complexes with 1,3,4,thia-diazole derivatives”. In this work, different approaches were used to characterize the prepared complexes and simulated them with theoretical programs. In vitro studies were performed to investigate the biological effect of the compound. Overall, the manuscript is well written, and I found this manuscript suitable to be published in its form.
the language is fine and easy to understand
Author Response
We thank the referee for carefully reading the paper and recommending it for publication.